# Brief Insights into mRNA Vaccines: Their Successful Production and Nanoformulation for Effective Response against COVID-19 and Their Potential Success for Influenza A and B

**DOI:** 10.3390/pathogens13060500

**Published:** 2024-06-12

**Authors:** Amerah Parveen, Amal Ali Elkordy

**Affiliations:** School of Pharmacy and Pharmaceutical Sciences, Faculty of Health Sciences and Wellbeing, University of Sunderland, Sunderland SR1 3SD, UK; amerahparveen.1991@gmail.com

**Keywords:** mRNA vaccines, vaccine carrier delivery systems, buffer choice, vaccine stability, lipid nanoparticles, COVID-19, influenza virus, seasonal influenza virus

## Abstract

A mRNA vaccine is a type of vaccine that induces an immune response. Antigen-encoding mRNA is delivered via vaccine carriers into the immune cells, which are produced because of antigen-encoding mRNA translation, a protein. For example, COVID-19 mRNA vaccines produce the spike protein of the COVID-19 virus, whereas for influenza virus, mRNA vaccines target the haemagglutinin protein to treat the flu, and it requires modifications depending on the pandemic or seasonal viruses as it is capable of adapting the immune response, which makes the development of vaccines arduous. The protein molecule promotes an adaptive immune response that eliminates and terminates the corresponding virus or pathogen. There are many challenges to delivering an mRNA vaccine into the body; hence, the encapsulation of the mRNA (usually within lipid nanoparticles) is necessary to protect the mRNA from the body’s surrounding environment. In this review article, we focus mainly on the production, formulation, and stabilization of mRNA vaccines in general, elaborating more on and focusing more on SARS-CoV-2, or COVID-19, and influenza viruses, which have become a major concern as these viruses have turned into life-threatening diseases.

## 1. Introduction

Ribonucleic acid (RNA) contains nucleotides consisting of ribose sugars attached to the nitrogenous bases and phosphate groups. RNA molecules consist of ribose sugars that make their shelf life short-expiry, whereas DNA consists of deoxyribose sugar-producing highly stable molecules [1]. RNA molecules help in regulating gene expression and also in curing diseases by acting as therapeutic agents. In protein synthesis, three types of RNAs are utilized, namely, messenger RNA (mRNA), transfer RNA (tRNA), and ribosomal RNA (rRNA). mRNA is transcribed from DNA containing the genetic information or genetic blueprint to form proteins. tRNA holds a role in translating mRNA into proteins, while rRNA forms ribosomes essential for protein synthesis [2].

mRNA transcripts are highly safe, have high transfection efficiency, and are less toxic as compared to DNA-based drugs. This is because simple mRNA does not enter the nucleus to execute its function, and there is no risk of insertional mutagenesis or any related infection. mRNA possesses a higher capability to treat diseases requiring protein expression and effective therapeutic efficiency because of continuous translation into encoded proteins, triggering long-lasting expression in comparison with transient peptides or protein drugs. These benefits of mRNA over DNA enable scientists to rapidly enter mRNA-based technology [3].

In this review article, we focus mainly on mRNA, recent advances in its stability when used as a vaccine, and technologies adopted to produce mRNA vaccines, for example, SARS-CoV-2 or COVID-19 vaccines. Moreover, we focus on the mRNA vaccines used for influenza A and B viruses, as they are capable of constantly mutating, which makes vaccines inefficient to perform in order to get rid of the flu.

## 2. Structure of mRNA

mRNA is a single-stranded ribonucleic acid macromolecule (typically ≈ 1000–4000 nucleotides) that can form a double-stranded molecule stabilized by intramolecular hydrogen bonds produced via the complementary pairing of contiguous bases, folding itself like a hairpin stem-loop or pseudoknots (Figure 1). mRNA is transcribed by a DNA strand (helping in protein synthesis) and provides coding information required for transcription and processing into functional proteins [1].

mRNA is divided into three parts [2] as follows: the nucleotide at the 5′ end serves as a binding site for proteins that promote polypeptide synthesis; the nucleotide in the middle signifies the sequence of amino acids in the polypeptide chain; and the nucleotide at the 3′ end regulates the mRNA’s stability (Figure 2). The details are discussed later.

mRNA contains specific information important for in vivo performance that can be lost if degradation occurs [5]. It contains a negative charge, which makes mRNA susceptible to ubiquitous RNases. Therefore, mRNA faces problems translating functional proteins in the cytoplasm as it cannot pass through the anionic cell membrane [6]. For each individual amino acid in proteins, there are three nucleotide units encoded, each consisting of a mass of around 330 Da. The 3-D structure of mRNA molecules (Figure 1) has emerged to understand the interrelationships between protein translational efficiency, higher order structure, and chemical stability [1].

Chemistry of mRNA and how degradation occurs:

RNA degradation depends mainly on how prone the molecule is to the in-line hydrolytic cleavage and attack by nucleases, chemical modifiers, and oxidizers in RNA. In-line hydrolytic cleavage is considered a universal intrinsic mechanism for RNA, among the other degradation processes. Cleavage of the RNA phosphodiester backbone bond is commenced by deprotonation of the 2′-hydroxyl group of the ribose molecule (Figure 2). The deprotonated OH group attacks the phosphate group to produce a pentacoordinate transition state. This transition state relies on the RNA backbone to adopt the confirmation, causing the oxyanion group to leave, which is in line with the 2′-OH group. This departure of the oxyanion group results in 2′,3′-cyclic phosphate, and the RNA strand breaks. Hydrolysis forms a fundamental limit for the stability of mRNA medicines and technologies [7].

## 3. mRNA Vaccine

During the pandemic, the vaccine was found to be the best alternative to protect human life from the deadly coronavirus. While researchers were trying to find ways to get rid of this virus, Chinese scientists in early January 2020 reported about the coronavirus RNA sequence. Scientists have achieved advancements in genetic-based vaccines and turned their efforts to creating novel vaccines to neutralize extremely contagious pathogens causing COVID-19 and influenza viruses.

The mRNA vaccine cannot integrate into the genomic sequence, which means that it is not susceptible to insertional mutagenesis. The first and foremost benefit of nucleic acid (DNA or RNA) vaccines is that they can transport several antigens at a single time, covering a wider range of tumor-associated antigens and augmenting the chances of bypassing vaccine resistance via the induction of a humoral or cellular immune response. Vaccines based on nucleic acid are less restricted by the human leukocyte antigen (HLA) class, induce larger T cells, and are capable enough to encode entire tumor antigens and allow the antigen presenting cells (APC) to present or cross-present various epitopes. They can encode a wide variety of antigens [8].

mRNA-based vaccinations have the benefits of eliciting robust T-cell immune reactions in addition to inducing humoral responses that impede viral growth. Furthermore, the amalgamation of distinct antigens, particularly those that are conserved, holds immense importance in the development of vaccines [9].

## 4. History of mRNA Vaccines

The mRNA vaccine is based on the principle that mRNA acts as an intermediate mediator after reaching the host cell through different routes that will translate into an antigen [10]. The mRNA molecule was invented in 1961, and its significance was discovered in 1990, when the gene mRNA was injected into mice for protein production [1].

In late 1980, delivering mRNA molecules was first tested by manipulating gene expression or attaining desired proteins in the cells. It was reported that effective mRNA transfection in NIH 3T3 fibroblasts with the help of cationic lipid N-(1-[2,3-dioleyloxy] propyl)-N, N, N-trimethylammonium chloride was possible. Over the past two decades, there have been many improvements and myriad technical developments with the idea of the expression of a particular gene by transfecting mRNA molecules in the host cells [10].

In 1990, the first testing was performed in an animal model by direct expression of external substances to have medicinal effects by transporting RNA vectors to encode a gene, for example, β-galactosidase and luciferase into murine muscle cells, and vasopressin mRNA was transfected into rats to reverse diabetes insipidus. In 1993, it was reported that synthesizing an in vitro mRNA vaccine that encodes the nucleoprotein of the influenza virus triggered the activation of cytotoxic T lymphocytes in vivo (using mice). Afterward, it was discovered that mRNA in vivo application produced both the activation of cytotoxic T cells and the humoral response of B cells responsible for producing specific antibodies [11]. Figure 3 depicts the key discoveries and advances in mRNA therapies.

Many researchers have carried out inventions for mRNA vaccines, which were applied to prevent lung carcinoma and triple-negative breast cancer using herpes simplex virus 1 thymidine kinase and mRNA’s encoding MUC1, respectively. mRNA vaccines have been used to recover from diseases like cancer, infectious diseases, allergies, and diseases needing protein replacement [10]. Table 1 reveals the advantages of mRNA vaccines.

## 5. How the mRNA Vaccine Works

The purpose of vaccines is to produce antibodies that help the body’s immune system fight against pathogens. For this, a dead form of the pathogen is injected to cause the immune system to build antibodies, and such vaccines are used against flu, rabies, hepatitis A, and polio. Another method is where a live virus is used, but in a weaker form of the pathogen that is unable to cause any illness or become contagious; these are live-attenuated vaccines used for treating measles, rubella, smallpox, mumps, and chicken pox [13].

mRNA vaccines use altogether different methods, where neither a live nor dead version is used. The drawback of mRNA vaccines is culturing and producing them on a large scale. mRNA vaccines work by producing an RNA copy synthetically of a required gene and introducing this to the somatic cells so that it can now act as mRNA. RNA is a sequence of nucleotides; the ribosome is unaware of whether the mRNA has arrived from the cell nucleus or was produced in vitro to be delivered into the body via injection [13]. After injecting mRNA, it gets transferred into the cell, where Poly A Binding Protein (PABP) binds to the poly-A tail and interacts with eukaryotic translation initiation factors (elfs) 4E [14,15]. The interaction of elfs with the 5′ cap, UTRs, PABP, initiator methionyl tRNA, and 40s ribosomal subunit renders the circularization of mRNA and initiation complex formation. After the 40s ribosomal subunit scans the transcription initiation codon, the 60s ribosomal subunit is employed, and elfs are freed to commence amino acid chain extension [3]. This is how mRNA is translated and coding is accomplished.

Normally, phagocytosis occurs on the antigen by the DC cells (dendritic cells), which will mature and migrate the antigens to the lymph nodes from around the body, allowing the antigen fragments on their surfaces to develop T cells. As a result, CD8 T cells mount an immediate immune response, and humoral immunity can neutralize infections if the same infection is produced in the future. B cells are stimulated to make antigen-specific antibodies through either direct recognition of the antigen at the B cell receptor or help from CD4 T cells. Before infection and proliferation, these antibodies can quickly develop an immune response upon recognition of the same antigen on the pathogen itself. The backbone of long-term vaccine-mediated immunity is this humoral protection [16].

In other words, mRNA vaccines trick the human body cells into producing the antigens of pathogens that the cells have never encountered. mRNA vaccines do not enter the cells or chromosomes, they just exploit the ready-made machinery of immune cells so that antibodies are produced to treat the infection and transmission [13]. Figure 4 shows how mRNA formulations work in the human body against infections, for example, SARS-CoV-2. The same principles apply to influenza viruses.

## 6. Challenges Associated with mRNA Vaccines

To achieve a robust vaccine, the successful design of the formulations, considering the physicochemical characteristics and conformational stability of mRNA, is important. The choice of the drug delivery system is a bigger challenge for mRNA as compared to proteins due to mRNA molecules being delivered intracellularly while proteins are restricted to targets outside the cells. Another problem is the intrinsic expeditious degradation of mRNA; however, it is quite favorable from pharmacokinetic and safety parameters, but unfortunately, it is more prone to deterioration [1]. Accordingly, mRNA should be shielded from proteolytic nuclease enzymes. Hence, lipid nanoparticle, LNP, formulation is proved feasible.

To overcome the challenges related to immunogenicity or to achieve prolonged stability and to obtain accurate and potent protein expression of the mRNA vaccines, chemical modifications, including cap structure and modification in nucleotides, play a key role. To acquire ideal target profiles for mRNA vaccines and related therapeutics, further innovative strategies are needed [1].

The new strategies that are being used in developing the vaccine discussed in this review are as follows: formulation of LNP via the microfluidic system, stabilizing mRNA, preventing degradation using different carrier molecules, for example, lipid-based delivery systems, polymer-based delivery carriers, peptide-based delivery systems, virus-like replicon particles (VRPs), and cationic nano-emulsions (CNE).

## 7. mRNA Stability

The stability of mRNA has been investigated both in vitro and in vivo. In vivo, stability is related to mRNA-inherent properties, which include optimization of regulatory regions, 5′cap, coding sequence, untranslated region (UTR), and poly-A tail of mRNA [17]. mRNA optimization is crucial for successful therapies.

mRNA has typically five functional regions: 5′ cap, 3′ Poly-A tail, open reading frame (ORF) flanking, and 5′ and 3′ untranslated regions (UTRs). Each region will be discussed to gain a better understanding of how the stability of mRNA can be improved.

Cap: The cap is located at the 5′ end of mRNA with different degrees of methylation. Specifically, 5′ cap (m^7^G ppp) is a single methylated nucleoside [18] and contains 7-methylguanosine (m^7^G), attaching the subsequent nucleotide with a 5′,5′- triphosphate bridge (ppp) in eukaryotes. (as shown in Figure 2). The cap binds to elfs4E through the hydrophobic cation interaction of m^7^G and a negative electrostatic charge of the triphosphate bridge during translation initiation [3].

The dithiodiphosphate modification introduced to the tri- or tetraphosphate bridge decreases the cap sensitivity to Dcp and increases RNA translation. Also, the phosphorothioate cap analogs increase the stability and translational efficiency of mRNA vaccines in immature dendritic cells. But phosphorothioate is position-sensitive and is associated with stereochemistry in catalysis. Recent studies have indicated that the 5′ cap structure is a major dominant way for the host to distinguish between self- or non-self mRNA molecules [10].

Poly-A Tail: It consists of 10 to 250 adenine ribonucleotide units. They play an important role as it form a dynamic addition to mRNA because their length helps in regulating the mRNA translation efficacy, stability, and protein expression by reducing the RNA exonuclease activity. Mechanically, the 3′ end of the Poly A tail combines with PABPs, which eventually interact with the 5′ cap via translation initiation factors elf4G and elf4E responsible for promoting closed-loop structure and increasing the affinity to the mRNA cap, which then regulates the translation efficiency of mRNA [3].

It was reported that even a short Poly A tail sequence of around 33 to 34 NT showed high translation efficiency [14] and it was also observed that at least 30–40 adenosines were deemed necessary to inhibit both the 5′ to 3′ and 3′ to 5′ mRNA degradation pathways [18].

5′ UTRs and 3′ UTRs: They do not participate in encoding the protein but are crucial for protein expression and for regulating the translation of mRNA. UTRs are involved in the subcellular localization of mRNA and in regulating the stability and translation efficiency of mRNA. Specifically, 5′ UTR is primarily involved in initiating the translation process, whereas 3′ UTR affects the stability and half-life of mRNA [3,19].

The internal ribosomal entry site at 5′UTR helps in recruiting ribosomes and initiates translation in cap and elf4E independently. The strongest Kozak sequence is widely used for improving protein translation. Importantly, the 3′ UTR contains an AU-rich zone that causes mRNA decay [18].

ORF: ORF has a crucial effect on the immunogenicity and translation efficiency of mRNA due to its recognition by cellular sensors [14]. It largely focuses on codon optimization and introducing functional peptides along with replication processes. The most used but controversial approach is codon optimization for improving the translation process. To protect mRNA from endoribonuclease degradation, increasing the guanine-cytosine (GC) content instead of replacing rare codons in ORF and enhancing protein expression of mRNA in vivo [20,21] has been used by CureVac for a recent SARS-CoV-2 mRNA vaccine candidate [14].

Additionally, the mRNA sequence dictates the secondary structure, which interestingly influences the degradation of mRNA by hydrolysis. A specially designed algorithm has reported that designs of optimal mRNA sequences for maximal base stacking regions have significantly enhanced mRNA stability [14].

For in vitro studies, the stability of mRNA formulations is related to nano-drug delivery systems such as lipid nanoparticle (LNP) size, mRNA integrity, lipid content, LNP delivery system composition, mRNA structure, and manufacturing processes. For guidance and advancements in vaccine development and quality control, regulatory establishments have issued guidelines and regulations for stability studies. Vaccine development can be achieved if mRNA vaccine degradation processes and factors influencing degradation studies are tackled.

In 2020, the Centre for Drug Evaluation (CDE) emphasized relevant guidelines for mRNA vaccine stability by considering conditions such as pH, humidity, light, temperature, and the number of freeze-thaw cycles. Followed by focusing on physicochemical properties and expression efficiency such as active ingredient contents, particle size, and mRNA encapsulation efficiency. WHO, in 2020, also supplied a guideline demonstrating that the stability of mRNA vaccines is affected by mRNA content, mRNA integrity, encapsulation efficiency, particle size, polydispersity, vaccine potency, and contaminants linked with lipids and mRNA. In 2022, CDE published technical rules for in vivo gene therapies, which mentioned nucleic acid-based products and specified that product design should take into consideration essentials, such as poly-A tail, 5′ cap, delivery systems, and nucleic acid modifications, which can affect the mRNA vaccine’s stability [17].

## 8. Manufacture of mRNA Therapies

The advancement of mRNA therapeutics majorly involves mRNA design, mRNA synthesis, pharmacology, safety parameters in vivo, manufacturing, and clinical trials. However, from the above steps, mRNA design, synthesis, and formulation hold the most crucial role in producing mRNA-based vaccines [3].

mRNA Design: Synthetic mRNA used for therapy is designed from the blueprint of eukaryotic mRNA and has five functional regions, which have been described earlier as they form the major role in the in vivo study for stability and the advances taking place. Cap and Poly-A tails present at the very 5′ and 3′ ends of mRNA are the most crucial elements. Cap and Poly A tail possess the main role in efficient translation and are also required to stabilize the mRNA in the cytosol, where decay is catalyzed predominantly by exonucleases (Figure 2 and Figure 5 present structural elements of mRNA regions). To increase translation and stability, mRNA further requires the support of 5′ and 3′ UTRs to flank ORF. UTRs are to be selected carefully because they can alter translation and stability [19].

mRNA synthesis: Functional synthetic mRNA is synthesized by the in vitro transcription (IVT) method using a cDNA template as the precursor, typically plasmid DNA (pDNA) using bacteriophage RNA polymerase (Figure 6). The first step to producing mRNA is the manufacturing of pDNA. While performing IVT, unpolished pDNA contains traces of bacterial genomic DNA and can get rid of by digestion with DNAses for purification [23].

The mRNA synthesis formed chemically displays an upper size chain limit of ~100 nucleotides after resulting in repetitive yields of >99%, whereas physiologically, mRNA typically comprises 1000–10,000 nucleotides. The length of mRNA is restricted when chemical synthesis is performed, while enzymatic IVT is an easy and inexpensive process for producing mRNA-yielding products in milligram quantities [22]. The manufacturing process utilizes Good Manufacturing Guidelines rather than using biologicals, which is likely to be advantageous [23]. Clones of pure and invariant pDNA are produced, which is the main aim of synthesizing an mRNA vaccine. Hence, the heterogeneous pDNA and remnants of bacterial DNA are not bothersome. However, the linear pDNA gets transcribed with the help of bacteriophage RNA polymerase because the unnecessary DNA will be eventually eliminated in further processing steps [19]. There are two ways to get the linearized DNA template for IVT: either amplify the target DNA in preclinical research or small-scale production using PCR, or linearize the plasmid using restriction enzymes [24].

IVT is dependent on RNA polymerases that are generated from bacteriophages. These polymerases enable robust transcription and make it easier to synthesize mRNA from chemically modified ribonucleoside triphosphates (rNTPs) with the aid of modified RNA polymerases [25].

Commercially, IVT is a highly efficient process using highly processive RNA bacteriophage single subunit polymerase, which is not only cost-effective with a higher yield but also easily scalable for producing mRNA (Figure 6 and Figure 7). Examples of bacteriophage RNA polymerases are T_3,_ T_7_, and SP_6_, which are single polypeptide chains only requiring Mg^+^ as a cofactor and run off the DNA template after various transcription processes. A desirable amount of pharmaceutical mRNA is produced in a cell-free environment by IVT encoding for the therapeutic protein or the antigen of the vaccine [22].

Compared to untreated mRNA, the translation of mRNA modified with 5-methylcytidine was four times greater [26].

The manufacturing of DNA has a few steps, as mentioned below: after the synthesis of DNA, the production of the desired mRNA takes place [22]. Figure 8 depicts the necessary steps for preparing in vitro transcription (IVT) of mRNA for the production of the mRNA drug substance. The steps include plasmid linearization, in vitro mRNA transcription, and the mRNA purification process; full details can be found in [22].

Purified mRNA is considered a naked form, which is not usually delivered as such. To effectively deliver the nucleic acid to the specified target cells and provide protection, a delivery vector, such as nanoparticles (NPs) to encapsulate mRNA, is used. The most extensively utilized vectors are lipid nanoparticles (LNPs) to encapsulate mRNA, as shown in Figure 7, which are also the ones employed and clinically used for SARS-CoV-2 vaccines [22], and they are under clinical trials for influenza vaccinations.

To briefly explain the immunological mechanism of mRNA vaccines, Figure 9 depicts the COVID-19 mRNA vaccine working mechanism as follows: after the intramuscular injection of the LNP-encapsulated mRNA vaccine is given, it enters muscle cells via endocytosis, and spike (S) protein is produced and then released out to the plasma membrane [10]. As discussed in a study by Park et al. [10], most of the S protein will be degraded in the proteasome (Figure 9) and merged within the class I major histocompatibility complexes (MHCs) and then presented to CD8+ and CD4+ T cells. Also, it was reported [10] that the major immunization mechanism for the mRNA vaccine is humoral immune response through B cell activation. The activated B cells will proliferate and differentiate into antibody-secreting plasma cells and memory B cells; this will take place once naïve B cells are activated by interacting with the associated CD4+ T cells and the ligation of CD40. Hence, by subsequent exposure to the antigen, the antibodies produced from plasma cells will neutralize the antigen and block the virus that carries the antigen from infecting its target cells [10].

## 9. General Factors Affecting mRNA Vaccine Quality and Stability

mRNA Carrier Delivery Systems: The delivery vehicles that are primarily used are LNPs, inorganic nanoparticle delivery systems, polymeric nanoparticle delivery systems, and peptide nanoparticle delivery systems, as shown in Figure 10. The LNPs protect mRNA from degradation by ribonucleases and enter the cell via endocytosis, eventually escaping into the cytoplasm with the help of endosomes to complete translation and expression of mRNA. For delivering mRNA to the site of action, LNP encapsulation helps increase the immunogenicity of mRNA vaccines. LNPs consist mainly of four main components: cholesterol (20–50%), phospholipids (10–20%), ionizable lipids (30–50%), and PEG lipids [17,28].

Polymer-based delivery system: (refer to Figure 10) Dendrimers, copolymers, and polyamines are examples in the literature of polymeric materials that help in delivering mRNA vaccines. They can protect mRNA from RNAse-related degradation like lipid-based systems, but polymers produce particles with a high polydispersity index. Cationic polymers, namely, polyamidoamine (PAMAM) dendrimer, polyethylenimine (PEI) (the most widely used polymeric material in vaccine delivery, as evident from the literature), and polysaccharide, help in delivering negatively charged RNA [29]. Yang et al. [26], in a review article, reported that combining the cationic cyclodextrin—a low-molecular-weight polymer—conjugates with mRNA for intranasal vaccination resulted in a strong immune response due to the high mucosal affinity of cyclodextrin and the adjuvant action of the cationic polymers.

Peptide-based mRNA delivery: (refer to Figure 10) It can be used as a primary carrier for RNA molecules only if it possesses a positive charge. The ratio of the amino group on a peptide (positively charged) to the phosphate group on RNA (negatively charged) affects the formation of nanocomplexes. Protamine is the only peptide carrier that has been evaluated in clinical trials [29]. Protamine is rich in arginine, has a small molecular weight, and contains nuclear localization signals. It can form stable complexes by self-assembling with negatively charged nucleotides. This complex not only protects from degradation but is also used as an adjuvant [26]. However, it has been demonstrated that mRNA can also be effectively stabilized against deterioration by serum components by complexation with protamine, a small arginine-rich nuclear protein that stabilizes DNA during spermatogenesis [19].

Virus replicon Particle: This carrier is used in SA-RNA, which self-replicates and expresses the required antigen efficiently. The viral structural protein is important for particle formation and is expressed in helper cell lines to package SA-RNA. The significance of this system is its efficiency in delivering RNA to the cytoplasm by viral vectors; this feature is attributed to the internalization of the viruses and the release of the genomes in the cells using different pathways effectively. The problem associated with viral replicon is the development of antibodies against viral vectors [29].

Cationic Nanoemulsion (CNE): It is a form of non-viral delivery system that works by binding to SA-RNA. The cationic lipid DOTAP (1,2-Dioleoyl-3-trimethylammonium propane) constitutes an essential component of CNE. The immune response induced by CNE is more robust in comparison to viral replicon protein. It is found to be beneficial when implemented against the Zika virus for protective immunogenicity [12]. Squalene-based CNE has been studied broadly, exploring the SA-RNA delivery system. CNE comprises an oily squalene core encircled by a cationic shell containing lipids, facilitating the stabilization and absorption of SA-RNA [30].

Lipid nanoparticles (LNPs)-based mRNA delivery system: This mRNA delivery system will be explained in more detail as LNPs are used clinically for COVID-19 vaccinations.

The ideal particle size of LNPs is 20–200 nm, and it influences their biodistribution, elimination, immunogenicity, internalization, and deterioration. Target delivery to particular tissues and cells can be achieved by adjusting the size of the LNP. The ideal size of the particle range makes LNP ideal for permeation and retention and allows easy crossing of interstitial tissues [31]. LNPs offer many benefits, such as flexibility in formulation and scaling up, a low toxicity profile, a high transfection capacity, modularity, compactivity with varying nucleic acid sizes, types, and protection of mRNA from internal deterioration, thereby extending the mRNA vaccine’s half-life [28].

Phospholipids, frequently employed as structural lipids, spontaneously unite to form bilayers of lipids that exhibit elevated phase transition temperatures. Phospholipid-1,2-dioleoyl-sn-glycero-3-phosphoethanolamine (DOPE) increases the effectiveness of mRNA transfection [32].

The chemistry of phospholipids can increase the delivery of mRNA via accelerating endosomal escape and membrane fusion [33]. Phospholipids are also called helper lipids as they provide structural stability to NPs as well as improve the biodistribution of LNP [34]. Phospholipid contains a phosphoethanolamine head, which increases endosomal escape because of its fusogenic property. Additionally, it was discovered that negatively charged phospholipids altered the LNP tropism from the liver to the spleen, whereas zwitterionic phospholipids mostly enhanced transport to the liver. Hence, depending on the choice of phospholipids incorporated, it will contribute intracellularly by enhancing the rate of endosomal escape. Furthermore, phospholipids regulate the transport of LNPs with selective organ targeting (SORT) to the spleen [33].

The LNP synthesized with various ionizable amino lipids influences how the phospholipids have an impact on the delivery of mRNA. C12-200 LNPs formulated using DOPE were better compared with DSPC (1,2-distearoyl-sn-glycero-3-phosphocholine)-formulated LNP. Therefore, phospholipids enhance the uptake of LNP, which enhances mRNA delivery to the cytosol. For effective transfection, the fusing of mRNA to lysosomes should be prevented; therefore, mRNA must escape the endosomes, and if fusion occurs, mRNA will be degraded. DSPC-LNP showed a higher coefficient than DOPE-LNP, signifying that DSPC LNPs are more frequently trapped in lysosomes and that mRNA cannot be sufficiently delivered into the cytosol [33].

Because mRNA is a single-stranded molecular structure with exposed nucleobases, it has a higher hydrophobicity and is, therefore, more likely to interact hydrophobically with liposomal lipids. The more compact complexes are produced when a higher proportion of DOPE and mRNA length are used. The charge ratio N/P (positively charged cationic liposomes: negatively charged ratio) delivering mRNA molecules is an influential factor that influences the effectiveness of the delivery of nucleic acid. The optimum value of the N/P ratio leads to the complete binding of RNA to lipoplexes and neutralizes the negatively charged RNA, resulting in protection from nuclease degradation. Increasing the amount of DOPE (zwitterionic lipid) to cationic amphiphile lipid 1,26-bis(cholest-5-en-3β0yloxycarbonylamino)-7,11,16,20-tetraazahexacosane tetrahydrochloride (2X3) led to an increase in zeta potential. From the analysis by atomic force microscopy (AFM), it was seen that at a N/P ratio of 10/1, which is the optimal ratio, both negatively charged nucleic acid molecules were bound completely while the positive charge was retained, an important aspect for cell penetration to be effective [35].

LNP3-mediated mRNA transfections with a slight negative charge and near-neutral charge led to significant efficiency in delivering mRNA and gene expression irrespective of cell type [36]. The particle size decreases during complexation, signifying that compact complexes are formed. This depends on the ratio of 2X3 and DOPE; increased DOPE concentration leads to increased particle size. The longer the mRNA molecules, the better the transfection effectiveness of the mRNA, the lesser the conformational modification, and the higher the molar proportion of DOPE found in a liposome. When it comes to the development of mRNA vaccines, 2X3 and DOPE can be used as efficient delivery systems [35].

Another ingredient that improves particle stability is cholesterol, which does so by modulating membrane permeability [37]. The biodistribution and effectiveness of LNP delivery can also be influenced by the molecular geometry of cholesterol and its derivatives [38]. Because of its analog with C-24 alkyl phytosterols, cholesterol has been shown to play a significant role in improving gene transfection and biodistribution of mRNA-LNPs, as per the studies reported by Jung et al. [34].

PEG lipids help to stabilize LNPs, regulate their size, and stop the particles from aggregating while they are being stored. In addition, it can delay the absorption of LNPs, reduce opsonization and elimination through cells of the reticuloendothelial system mediated by serum proteins, and prolong the half-life of LNPs in circulation. Furthermore, PEG lipid functional changes can make it easier for LNPs to bind to ligands or other biomolecules [17].

Ionizable lipids, which can be single or multi-charged, have tertiary amine moieties and are neutral at physiological pH when they are found in the center of LNPs. However, in the acidic environment of endosomes, the ionizable group of the lipid becomes positively charged, which enhances endosomal escape and hence mRNA release into the cytoplasm [39]. An ionizable head group, a linker region, and the hydrocarbon chains make up ionizable lipids [14].

pH-sensitive ionizable lipids are more advantageous for in vivo mRNA distribution because neutral lipids associate minimally with the anionic membrane of vascular cells and significantly increase the biocompatibility of LNPs [17]. This further aids in preventing anionic biomolecules from binding non-specifically [40]. Ionizable lipids can escape from endosomes, and it was found that these lipids affect the size of LNP particles and are crucial in determining the natural adjuvant action of LNPs [40]. One of the keys to unlocking LNPs’ full potential is increasing the proportion of RNA released into the cytosol [34].

Phospholipids, which improve endosomal escape and fusogenicity; cationic-ionizable amino lipids, which condense with nucleic acid at low pH; polyethylene glycol (PEG) lipids, which provide steric stabilization of the formulation prior to use; and cholesterols, which allow vesicle stability both in vivo and in packaging, vials, are all used in LNP formulation [34].

COVID-19 mRNA vaccines from Pfizer/BioNTech and Moderna contain single-charged ionizing lipids. The immunological efficacy of mRNA vaccines is enhanced by multi-charged lipids because they have a greater proportion of positively charged polymeric amino (nitrogen) units than negatively charged nucleic acid phosphate moieties. This makes them a better candidate for mRNA encapsulation, lysosomal escape, and cellular absorption [17].

## 10. Factors Affecting mRNA-LNP Vaccine Stability

### 10.1. Physicochemical Properties

The integrity of the mRNA molecule is related to the mRNA vaccine’s stability and efficacy, which is a critical quality attribute (CQA), including UTRs, 5′cap, poly-A tail, and coding region. The removal of mRNA fragments has resulted in significantly reducing the duration of action of mRNA, causing deterioration and translation error, and hence, the length of mRNA fragments also impacts their stability [17].

The biological system’s interaction process, which includes dispersion, response, internalization, deterioration, and elimination, is determined by size and surface area. Decreasing the size increases the surface area, making it more reactive to the biological milieu. The ideal particle size is ~50 nm for highly efficient mRNA delivery, irrespective of chemical constituents. Charge depicts the fate of biodistribution and the efficacy of mRNA LNP. The negative charge of RNA interacts with positively charged cationic lipids, leading to successful encapsulation. Shape and internal structure influence cellular uptake and interaction with the surrounding biological system. Surface composition can be improved by incorporating PEG lipids into LNP by PEGylation-altering nanocarrier trafficking, resulting in enhanced shelf life [28].

The length of mRNA is measured in terms of several nucleotides. The length of the RNA vaccine is inversely proportional because there are more sites available for degradation with an increased number of nucleotides. In simple terms, mRNA length harms the shelf life of mRNA [41]. VAX-sequence is a quantitative method used to calculate the length of the mRNA vaccine and measure the mRNA’s integrity. Longer read sequences provide a quantitative profile of the integrity of mRNA vaccines during manufacture, storage, or transportation, whereas short sequences display inefficient and uneven coverage and prevent the analysis of the length and integrity of mRNA [42]. LNP3-mediated RNA with length (>1000 nt) transfection resulted in poor efficiency [36].

### 10.2. Excipients and Buffer System

Undoubtedly, RNase-free excipients are required for mRNA-based vaccinations [43]. Buffer systems and pH are important parameters to be taken care of during the formulation of nanoparticles, as pH affects the rate of hydrolysis in mRNA and is hence responsible for mRNA vaccine stability. mRNA is found to be more stable in a very mild alkaline environment [43]. The buffer utilized during the mRNA transmission plays a crucial role in regulating the transfection ability in vivo and in vitro and the assembly efficiency of mRNA-LNP, which further affects the expression level of related genes. It also regulates the biological distribution of LNPs and the accumulation of NPs non-specifically. Therefore, it is pivotal to consider the properties of the buffer and to understand the interaction occurring biologically in NPs to use as a guide for delivering mRNA LNPs [36].

The approved COVID-19 vaccines Moderna and Pfizer/BioNTech had the pH controlled between 7 and 8. The hydrolysis of nucleic acid was significantly accelerated by reducing the pH from 7 to 6.5. The phosphodiester bonds present in RNA molecules become more susceptible to hydrolytic breakage in the presence of Mg^2+^ or Ca^2+^. For the vaccine buffer system, a suitable buffer system and osmolarity regulator are needed due to the possibility of storing the mRNA vaccine below 0 °C, and after freezing, the pH may be altered. Initially, Pfizer used phosphate buffer consisting of NaCl/KCl in the COVID-19 vaccine. Later, phosphate buffer was replaced with Tris buffer due to the fluctuation in pH in the phosphate system after freezing [17]. The selection of a proper buffer system and osmolarity are quite important as the vaccine will be stored at a freezing temperature, which will eventually affect the pH. For example, sodium-phosphate buffer. Tris HCL stabilizes nucleic acid molecules and scavenges hydroxy radicals [43,44].

It is currently acknowledged that after mRNA-LNP formulation, cryoprotectants, like sucrose or trehalose, should be added in order to freeze LNPs [14].

### 10.3. Manufacturing Processes and Storage of LNPs

There are many methods to produce lipid-based nanoparticulate systems, such as ethanol injection and microfluidic technology. Microfluidic mixing is the most effective process.

Microfluidic Mixing: The microchannel operating conditions consist of laminar flow with slow molecular diffusion via channels. The development of structured patterns of microfluidic chips has led to improvements in homogenous particle size, its distribution, and better encapsulation in comparison to bulk procedures. The rapid mixing results from the staggered herringbone structure, which escalates the polarity of a mixture, leading to supersaturation and hence the formation of LNP. It can generate over a 100-fold production rate as compared to a single channel of a microfluidic chip [34].

Shelf life, temperature, and storage: One of the most important factors that has seriously affected the vaccine’s stability, and the major concern is protecting against structural alteration. It was revealed that there was some degree of degradation at different temperatures (−20, 5, 25, and 40 °C), with a significantly high percentage of loss of mRNA integrity with increasing ambient temperature. From the data of 2 COVID-19 vaccines, it has been experienced that Moderna was stable at −20 °C for 6 months, whereas Pfizer vaccine was stable at −80 to −60°C for 6 months, but when both were stored at 2–8 °C, stability or shelf life dropped to 30 days for Moderna and 5 days for Pfizer vaccines. The key factor in improving the stability of the vaccine is the mRNA structure itself. Strict control of temperature and rational process design are crucial for temperature-sensitive mRNA vaccines and their stability [17].

## 11. Application of the mRNA Vaccine in COVID-19

Rapid mutations occur in COVID-19 during transmission, and reports of alterations with possible immune escape have been found. These mutations can provide the virus with epidemiological, immunological, or etiological properties. The worldwide COVID-19 public health emergency must be controlled promptly with the use of a SARS-CoV-2 vaccination. Numerous COVID-19 mRNA vaccines are starting trials in humans [26]. Table 2 depicts the registered clinical trials (completed and in progress) at ClinicalTrials.gov, as of 30 May 2024, for lipid nanoparticle mRNA vaccine candidates against COVID-19. 

When self-amplified mRNA was compared to non-replicating mRNA, self-amplified mRNA varies as its full length is larger and its ORF not only encodes the target antigen but also produces a viral replication mechanism that leads to intracellular RNA to self-amplify and increases protein expression [26].

Moderna and Pfizer’s vaccines are based on mRNA technology, consisting of a single strand encapsulated in solid LNPs, which is administered intramuscularly. They specifically utilize nucleoside-modified mRNA, which makes it capable of reducing its inherent immunogenicity. They encode the mutated viral spike glycoprotein used by the SARS-CoV-2 virus to enter the cells. These two vaccines have been engineered in the same way, with slight differences in their composition of ionizable cation lipid and PEG lipid. PEG-lipids affect myriad criteria of solid LNP vaccines and play a role in RNA encapsulation efficiency and pharmacodynamics and pharmacokinetic properties, which include in vivo distribution, transfection efficiency, circulation half-life, immune response, and stability of the product. SLNs (solid lipid nanoparticles) are thought to be less toxic because they consist of natural lipids acting as carriers as well as being pharmacologically inactive. SLNs have various advantages, among which the most important is that they protect the mRNA from degradation by ribonucleases, which prolongs the blood circulation time of the vaccine and consequently reduces the clearance mediated through the kidneys and the phagocytic system. For the selection of organs, lipid concentrations can be varied to achieve the drug reaching its desired location. Moderna and Pfizer vaccines were labeled with DiD fluorophore indicators to investigate the endosomal escape process and cell internalization process and identify the differences concerning their physicochemical properties [45]. Table 3 depicts a collection of observations.

The BNT162b1 and BNT162b2 vaccines contain nucleoside-modified mRNA, which improves mRNA translation by substituting N1-methylpseudouridine with all uridines [46].

Clinical study results indicate that mRNA-1273 is typically well tolerated and safe for treatment in both adults and adolescents. and so far, no significant safety issues have been found [24]. Corbett et al. [47] documented that in preclinical models, the administration of mRNA-1273 elicited strong humoral and cellular immunity against both the original and mutant SARS-CoV-2 (D614G).

The immunization with 15 μg of ARCoV significantly produced neutralizing geometric mean titers that were roughly twice as high as those from the convalescent sera of COVID-19 patients, indicating a potentially effective treatment against COVID-19 [48]. CureVac has created RNActive^®^ vaccines, which use naturally occurring nucleotides complexed with protamine to exhibit inherent self-adjuvant action [46]. It has shown 47% protective action against COVID-19 and is also the lowest protection feature amongst all the vaccines whose clinical trials have been performed to date [26].

## 12. Application of the mRNA Vaccine for the Influenza Virus

When comparing several delivery methods for delivering naked mRNA vaccines, it was found that intranodal or near-nodal (into soft tissue) injections produced substantial immunogenicity against influenza A hemagglutinin virus and ovalbumin; this was observed after giving numerous injections into the lymph nodes of mice [19,49].

At the moment, influenza B viruses (IBVs) belonging to one or two antigenically distinct lineages, along with H1N1 and H3N2 influenza A viruses (IAVs), circulate periodically in the human population. Although research has boosted our understanding of pandemic risk, the exact influenza subtype that will start the next pandemic remains unpredictable. A small set of antigens with conserved epitopes found in several influenza virus subtypes are included in most universal influenza vaccines. A different strategy to achieve universal protection is to create multivalent vaccinations that contain antigens from all recognized subtypes of the influenza virus [50].

Three self-amplifying mRNAs producing hemagglutinin (HA) from three distinct influenza virus strains (H1N1, H3N2 (X31), and B (Massachusetts)) were created by a medium-length PEI in a similar mass, co-delivered to mice intramuscularly, which protected mice against viral challenge to widen immunity with a multivalent mRNA vaccine [51] via [29].

RNA-LNP vaccines modified with nucleosides have become an effective vaccination platform for the management of infectious illnesses. The study shows the distinct advantages of the pentavalent mRNA-LNP vaccine over monovalent constructs and generates widely protective immune responses, laying the groundwork for future examination of this immunization regimen [52].

For example, after viral exposure, NP-specific CTLs can quickly multiply, distinguish, and travel to infection sites. Additional information on the fundamental process driving cellular immune responses was obtained using single-cell RNA-seq analysis. Furthermore, there was a significant increase in the expression levels of genes linked to chemokine ligands (such as Ccl3, Ccl4, and Ccl5), chemokine receptors (such as Cxcr6), and granzyme (such as Gzmk and Gzma) in CD8+ T cells. These findings suggested that the stimulation and cytotoxic activity of T cells were responsible for the enhancement of protection activity [9].

The very first mRNA vaccine for preventing the spread of influenza A virus (H3N2) to enter clinical trials was recently revealed by Sanofi and Translate Bio, which stated that they had started a phase I clinical trial of an mRNA vaccine against seasonal influenza [30]. Five mRNA vaccines for influenza viruses that encode the HA antigen are currently undergoing clinical trials: Moderna’s mRNA-1851, mRNA-1440, and mRNA-1010, while Sanofi and Translate Bio have co-developed MRT-5400 and MRT-5401 [24]. Table 4 summarizes the registered clinical trials (completed and in progress) at ClinicalTrials.gov, as of 30 May 2024, for mRNA vaccines against influenza. Interestingly, some of these vaccines are combinational vaccines against influenza and COVID-19, for example, refer to study NCT number: NCT06097273 in Table 4.

The relatively poor immunogenicity of the H7N9 HA protein is indicated by the relatively small immune responses to unadjuvanted vaccinations and the low HAI titers found in naturally occurring infections. Adjuvantation was found to be needed for several H7N9 vaccine candidates to achieve satisfactory rates of seroconversion and seroprotection. HAI geometric mean antibody titers (GMTs) and seroconversion rates for these candidates were low in the absence of adjuvant (40–47% seroconversion, GMTs 24.1–32.8). Furthermore, six months after immunization, the H7N9 mRNA vaccine demonstrated measurable and persistent HAI titers, implying the progress of memory B-cell reactions. The initial mRNA vaccinations against the influenza viruses H7N9 and H10N8 produced strong humoral immune responses and were well tolerated [53].

There are many influenza vaccines, including inactivated virus, live attenuated virus, and recombinant haemagglutinin (HA), that target the haemagglutinin (HA) protein responsible for entry of the virus into the host. But the virus is still able to cause respiratory illness because of virus mutations. Hence, mRNA vaccines are under clinical trials [12]. Table 4 depicts clinical trials for mRNA vaccines against influenza, as registered at ClinicalTrials.gov as of 30 May 2024.

The seasonal influenza virus epidemic is a major threat to human health. Live attenuated and inactivated vaccines have resulted in restricted effects, leading to reformulation to meet the demand for the needs related to viruses. The vaccines induce immune antibody responses to fight against the immunodominant globular head domain of the virus hemagglutinin (HA); however, the virus is capable of escaping from the immune response owing to the plasticity of the HA head. Alternatively, a broader spectrum can be induced for protection that would target specific viral regions, for instance, the stalk domain of HA, which is less tolerant of escape [54].

Based on the above studies, there will be a potential for flu vaccines to be in the form of mRNA to be used in clinics.

## 13. Side Effects of mRNA-Based Vaccines

mRNA-LNP-based vaccines for COVID-19 have saved the lives of millions of people all over the world. Despite this fact, vaccines are similar to all other medicines in producing side effects in some people. Some of the side effects have been determined during the clinical trials or reported after approval and administration of those vaccines. Side effects include but are not limited to: injection site pain, headache, fatigue, myalgia, diarrhea, lymphadenopathy, and injection site redness and swelling (emc [55], accessed on 5 June 2024). Myocarditis and other cardiovascular conditions such as thrombocytopenia and thrombosis have also been reported in some mRNA vaccine recipients [56]. Hence, any serious side effects need to be taken into consideration by researchers and vaccine providers for the better safety of vaccines’ recipients.

## 14. Conclusions and Future Prospective

The advances in pharmaceutical technology and in cell biology have led to the in vitro translation of mRNA, which produced a pure drug substance to be encapsulated into nanoparticles for prophylactic infection with serious pathogens such as viruses. The mRNA-vaccine-based technology can be adapted and manufactured in large-scale production. The COVID-19 mRNA-based vaccine generally paved the way for mRNA-based vaccines against influenza, as confirmed by the considerable number of influenza mRNA vaccines that undergo clinical trials.

The challenges with mRNA vaccine stability need to be addressed, not only during downstream production but also after manufacturing the medicinal products to extend their shelf lives. For example, drying methods can be applied to produce dried powder vaccines to be reconstituted before administration. LNPs in clinics with ionizable lipids are the most successful carrier system for the mRNA as a medicinal product; hence, the discovery of more novel carrier-stable systems will be worthwhile, and this may help to overcome some of the side effects associated with the currently available mRNA COVID-19 vaccines.

## Figures and Tables

**Figure 1 pathogens-13-00500-f001:**
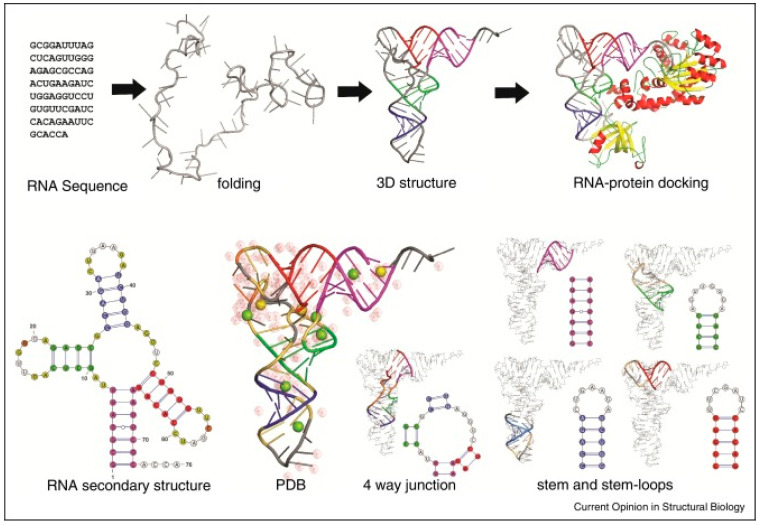
A summary of RNA folding and 3D RNA structure. (Top) A cartoon of the modeling process where a sequence is transformed from an unfolded structure to the native structure and further changed by docking with proteins. (Bottom) An example of the conceptual flow from RNA secondary structure diagrams to the corresponding 3D RNA structure, including the interaction of this 3D structure and its local motifs with ions and water ([4], under the terms of the Creative Commons license, http://creativecommons.org/licenses/by/4.0/ accessed on 30 May 2024).

**Figure 2 pathogens-13-00500-f002:**
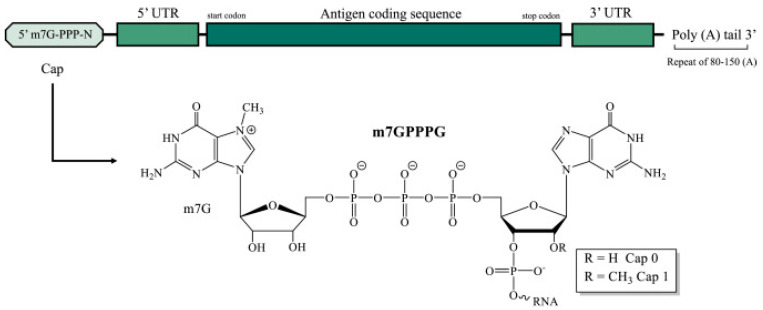
Schematic diagram of IVT mRNA primary structure (5′ terminal cap, untranslated regions, antigen coding region, 3′ poly-A tail), including chemical structure of the 5′-cap dinucleotide caps. (UTR, untranslated regions) ([5], under the terms of the Creative Commons license, http://creativecommons.org/licenses/by/4.0/ accessed on 30 May 2024).

**Figure 3 pathogens-13-00500-f003:**
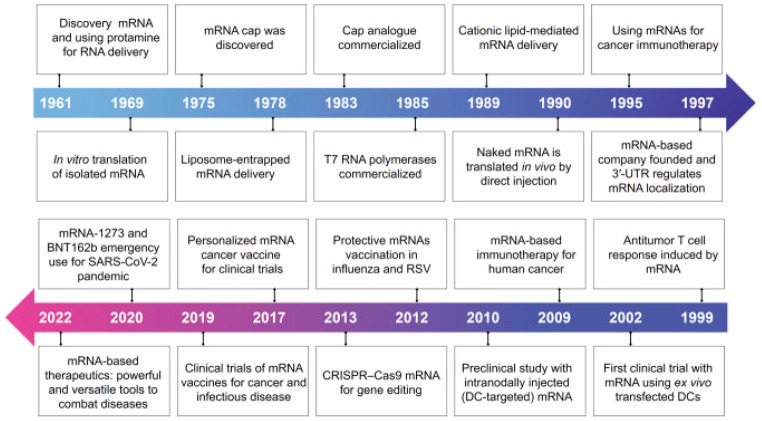
Key discoveries and advances in mRNA-based therapeutics ([3], under the terms of the Creative Commons license, http://creativecommons.org/licenses/by/4.0/ accessed on 30 May 2024).

**Figure 4 pathogens-13-00500-f004:**
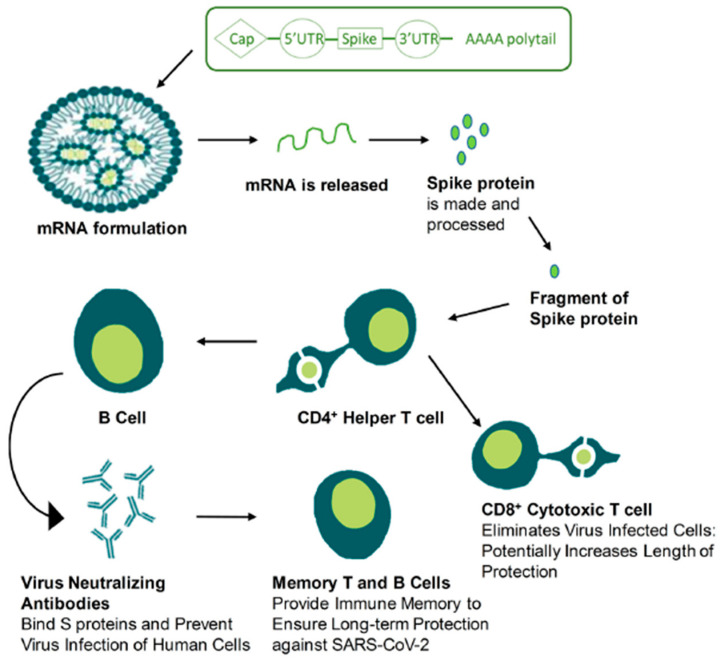
How mRNA vaccines work—training the immune system for a real infection (Taken with permission from Hussain et al. [12]).

**Figure 5 pathogens-13-00500-f005:**
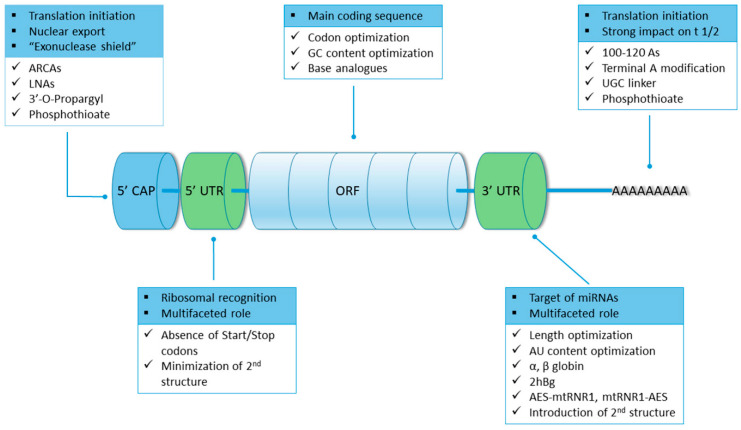
Structural elements of mRNA, their role, and modifications impacting mRNA viability as a therapeutic. The main role of each element (5′ cap, 5′ and 3′ UTRs, ORF, and Poly(A) tail) is given by a square bullet, whereas the modifications pertaining to its modification are given with checkmarks ([22], under the terms of the Creative Commons CC BY license).

**Figure 6 pathogens-13-00500-f006:**
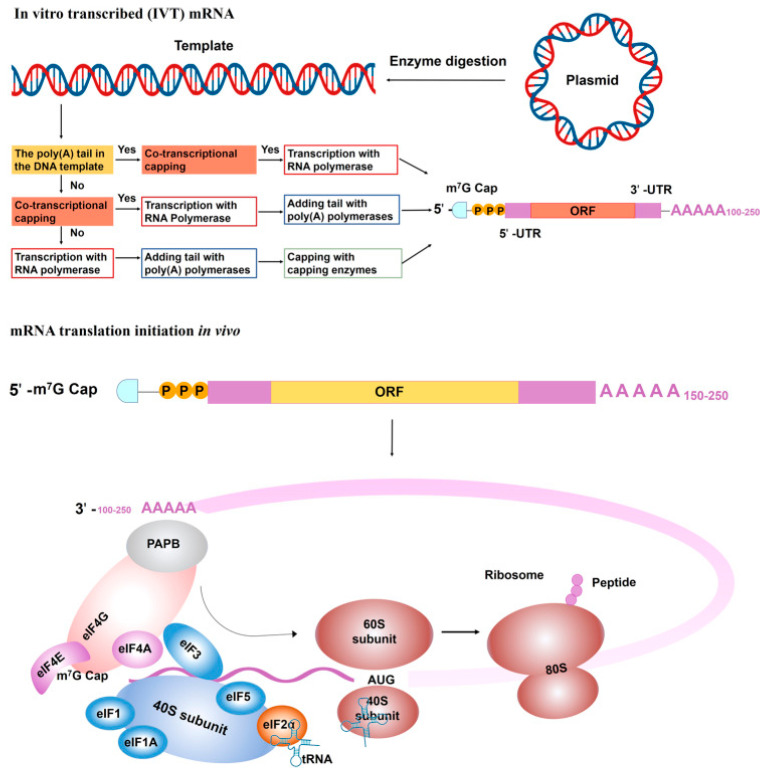
In vitro transcribed (IVT) mRNA and translation initiation ([3], under the terms of the Creative Commons license, http://creativecommons.org/licenses/by/4.0/ accessed on 30 May 2024).

**Figure 7 pathogens-13-00500-f007:**
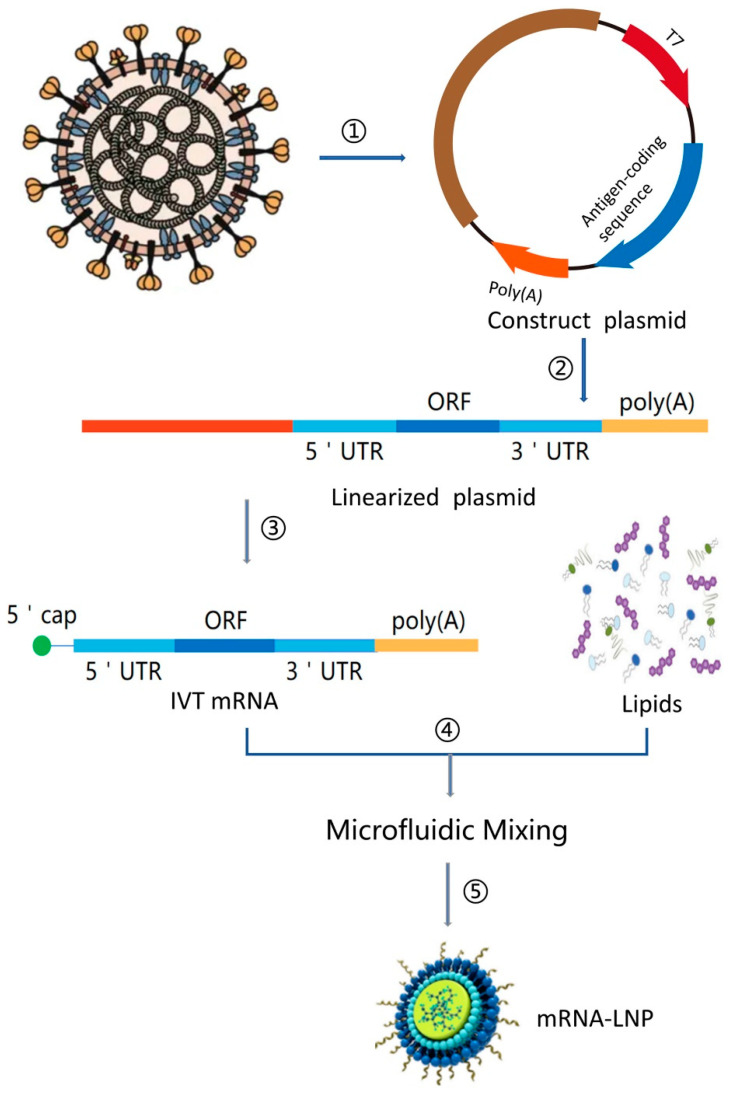
mRNA vaccine manufacturing process. (1) Sequencing and analysis of essential proteins of the virus, (2) the introduction of the plasmid into Escherichia coli and cultured and proliferated, (3) plasmid extraction, purification, and enzymatic digestion; (4) in vitro transcription of mRNA; (5) microfluidic mixing, and (6) the process of encapsulating mRNA into lipid nanoparticles (LNPs) ([26], under the Creative Commons Attribution license (CC-BY)).

**Figure 8 pathogens-13-00500-f008:**
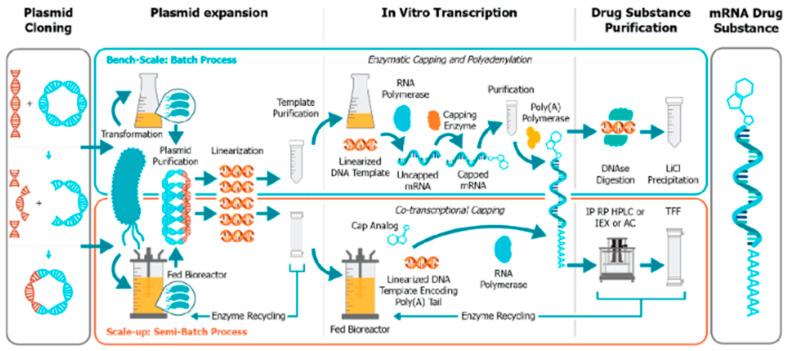
Process of IVT and DS purification and upstream processes to produce the DNA template. Acronyms: IP RP HPLC = ion-pair reverse phased high-performance liquid chromatography; IEX = ion exchange chromatography; AC = affinity chromatography; TFF = tangential flow filtration ([25], under the CC BY-NC-ND 4.0).

**Figure 9 pathogens-13-00500-f009:**
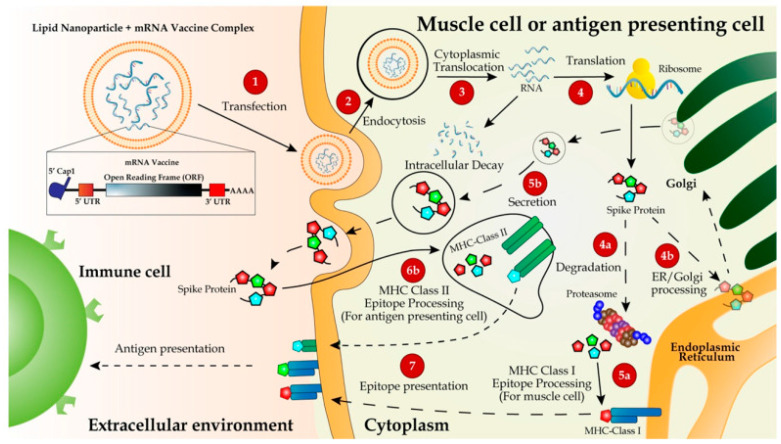
Delivery and working mechanism of a mRNA vaccine. mRNA vaccine, containing the coding region of S protein flanked by the optimized 5’- and 3’-UTRs and polyA tail, is synthesized via IVT, followed by 5’-capping with a 5’-cap analogy and encapsulation with LNP for IM injection (step 1). The vaccine is delivered into muscle cells or antigen-presenting cells such as dendritic cells or macrophages via endocytosis (step 2). mRNA molecules are unloaded from LNPs and translated to S proteins in the ribosome (step 3). The newly synthesized S protein is secreted into extracellular space, internalized via endocytosis into antigen-presenting cells, and incorporated as a part of the MHC class II antigen presentation complex (steps 5b, 6b, and 7) to present the antigen to immune cells, including T and B cells [27]. Partially degraded S peptides by proteosomes are incorporated into MHC class I complexes, which are then transported to plasma membranes and also presented as antigens to immune cells (steps 4a, 4b, 5a, and 7) ([10], under the Creative Commons Attribution License, https://creativecommons.org/licenses/by/4.0/ accessed on 30 May 2024).

**Figure 10 pathogens-13-00500-f010:**
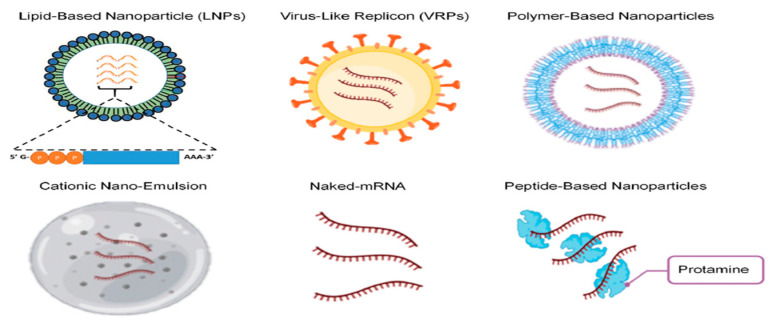
Various delivery technologies for mRNA vaccines are shown: lipid-based particles, virus-like replicon particles, polymer-based delivery, cationic nanoemulsions, naked mRNAs, and peptide-based delivery (taken with permission from Hussain et al. [12]).

**Table 1 pathogens-13-00500-t001:** Advantages and disadvantages of mRNA vaccines (information was taken from Hussain et al. [12]).

Sr. No	Advantages	Disadvantages
1.	Requires a shorter time to develop	Instability due to enzymatic degradation by natural RNases inside the body
2.	In vitro transcribed reaction is easy and quick, and manufacturing produces a high yield	There are myriad nucleotides in mRNA, which has to reach the cytosol at full length for translation to occur. Due to degradation, delivery of mRNA is a major concern
3.	Protein purification is not needed due to the in-situ synthesis of antigen-protein stability	
4.	If protection against RNases is ensured, there will be low degradation, and storage and transportation of mRNA might become easier as compared to protein-based vaccines	
5.	Safety and efficacy	

**Table 2 pathogens-13-00500-t002:** Clinical trials for mRNA-Lipid nanoparticle (mRNA-LNP) vaccines against COVID-19 (as registered at ClinicalTrials.gov, accessed on 30 May 2024). Full information on each vaccine, including the sponsor and the outcomes, can be found by referring to the study URL as per the table.

NCT Number	Study Title	Study URL	Study Status	Interventions, All Biologicals, Unless Stated Otherwise	Phases	Enrollment
NCT05534048	Booster Study of PTX-COVID19-B in Adults Aged 18 Years and Older	https://clinicaltrials.gov/study/NCT05534048 (accessed on 30 May 2024)	Not yet recruiting	PTX-COVID19-B|ComirnatyÂ^®^	PHASE3	3800
NCT05534035	Booster Superiority Study of PTX-COVID19-B Compared to VaxzevriaÂ^®^ in Adults Aged 18 Years and Older	https://clinicaltrials.gov/study/NCT05534035 (accessed on 30 May 2024)	Not yet recruiting	PTX-COVID19-B|VaxzevriaÂ^®^	PHASE3	450
NCT04785144	Safety and Immunogenicity Study of a SARS-CoV-2 (COVID-19) Variant Vaccine (mRNA-1273.351) in NaÃ¯ve and Previously Vaccinated Adults	https://clinicaltrials.gov/study/NCT04785144 (accessed on 30 May 2024)	Completed	mRNA-1273|mRNA-1273.351	PHASE1	135
NCT04566276	ChulaCov19 Vaccine in Healthy Adults	https://clinicaltrials.gov/study/NCT04566276 (accessed on 30 May 2024)	Completed	ChulaCov19 vaccine|OTHER: Placebo	PHASE1|PHASE2	192
NCT04811664	A Study of SARS-CoV-2 Infection and Potential Transmission in Individuals Immunized With Moderna COVID-19 Vaccine	https://clinicaltrials.gov/study/NCT04811664 (accessed on 30 May 2024)	Completed	Moderna COVID-19 Vaccine	PHASE3	1923
NCT05876364	Study to Assess Safety, Reactogenicity and Immunogenicity of the repRNA(QTP104) Vaccine Against SARS-CoV-2 (COVID-19)	https://clinicaltrials.gov/study/NCT05876364 (accessed on 30 May 2024)	Active, not recruiting	QTP104 1 µg|QTP104 5 µg|QTP104 25 µg	PHASE1	36
NCT04847102	A Phase III Clinical Study of a SARS-CoV-2 Messenger Ribonucleic Acid (mRNA) Vaccine Candidate Against COVID-19 in Population Aged 18 Years and Above	https://clinicaltrials.gov/study/NCT04847102 (accessed on 30 May 2024)	Unknown	SARS-CoV-2 mRNA Vaccine|Placebo	PHASE3	28,000
NCT05525208	Booster Study of COVID-19 Protein Subunit Recombinant Vaccine	https://clinicaltrials.gov/study/NCT05525208 (accessed on 30 May 2024)	Recruiting	SARS-CoV-2 subunit protein recombinant vaccine|Active Comparator	PHASE2	900
NCT05057169	Randomized Trial of COVID-19 Booster Vaccinations (Cobovax Study)	https://clinicaltrials.gov/study/NCT05057169 (accessed on 30 May 2024)	Active, not recruiting	BNT162b2|CoronaVac	PHASE4	400
NCT05168813	Efficacy Study of COVID-19 mRNA Vaccine in Regions With SARS-CoV-2 Variants of Concern	https://clinicaltrials.gov/study/NCT05168813 (accessed on 30 May 2024)	Completed	Moderna mRNA-1273|Vaccine 3 Dose|Vaccine 2 Dose	PHASE2|PHASE3	14,232
NCT04776317	Chimpanzee Adenovirus and Self-Amplifying mRNA Prime-Boost Prophylactic Vaccines Against SARS-CoV-2 in Healthy Adults	https://clinicaltrials.gov/study/NCT04776317 (accessed on 30 May 2024)	Completed	ChAdV68-S|ChAdV68-S-TCE|SAM-LNP-S|SAM-LNP-S-TCE|OTHER: Sodium Chloride, 0.9%	PHASE1	81
NCT05658523	COVID-19 Booster Study in Healthy Adults in Australia	https://clinicaltrials.gov/study/NCT05658523 (accessed on 30 May 2024)	Active, not recruiting	Bivalent Moderna|Novavax	PHASE3	497
NCT04283461	Safety and Immunogenicity Study of 2019-nCoV Vaccine (mRNA-1273) for Prophylaxis of SARS-CoV-2 Infection (COVID-19)	https://clinicaltrials.gov/study/NCT04283461 (accessed on 30 May 2024)	Completed	mRNA-1273	PHASE1	120
NCT04889209	Delayed Heterologous SARS-CoV-2 Vaccine Dosing (Boost) After Receipt of EUA Vaccines	https://clinicaltrials.gov/study/NCT04889209 (accessed on 30 May 2024)	Completed	Ad26.COV2.S|BNT162b2|mRNA-1273|mRNA-1273.211|mRNA-1273.222|SARS-CoV-2 rS/M1	PHASE1|PHASE2	867
NCT05516459	Prospective Monitoring of BNT162b2 Second Vaccination Booster Effects in Health Care Workers (HCW)	https://clinicaltrials.gov/study/NCT05516459 (accessed on 30 May 2024)	Active, not recruiting	Pfizer BNT162b2 Vaccine		635
NCT05289037	COVID-19 Variant Immunologic Landscape Trial (COVAIL Trial)	https://clinicaltrials.gov/study/NCT05289037 (accessed on 30 May 2024)	Completed	DRUG: AS03|BNT162b2|BNT162b2 (B.1.1.529)|BNT162b2 (B.1.351)|BNT162b2 bivalent (wildtype and Omicron BA.1)|BNT162b2 bivalent (wildtype and Omicron BA.4/BA.5)|CoV2 preS dTM [B.1.351]|CoV2 preS dTM/D614|CoV2 preS dTM/D614 + B.1.351|mRNA-1273|mRNA-1273.351|mRNA-1273.529|mRNA-1273.617.2|OTHER: Sodium Chloride, 0.9%	PHASE2	1270
NCT05057182	Third Dose of mRNA Vaccination to Boost COVID-19 Immunity (mBoost Study)	https://clinicaltrials.gov/study/NCT05057182 (accessed on 30 May 2024)	Active, not recruiting	BNT162b2	PHASE4	300

**Table 3 pathogens-13-00500-t003:** Tests performed to check the stability and physicochemical properties of Moderna and Pfizer COVID-19 mRNA-LNP vaccines [45].

Sr. No.	Contents and Tests Performed	Moderna	Pfizer	Results
1.	Ionizable Lipid	heptadecan-9-yl 8-{(2-hydroxyethyl) [6-oxo-6-(undecyloxy) hexyl] amino} octanoate (SM-102)	[(4-hydroxybutyl) azanediyl] di(hexane-6,1-diyl) bis(2-hexyldecanoate) (ALC-0315)	
2.	The molar ratio of cationic lipid:PEG-lipid:cholesterol:DSPC	50:1.5:38.5:10	46.3:1.6:42.7:9.4	
3.	DLS	93.00 ± 3.00 nm	92.89 ± 2.53 nm	Similar hydrodynamic size
4.	CD spectra in the far UV region	Resembles the same as standard mRNA. No change	Same as Moderna	No substantial change in structure and concentration implies SLN has a protective role.
5.	RP-HPLC (205 nm signal)	Retention time of 13.6 min	Retention time is 13.6 min	Both showed the same peak. Helps in calculating lipid content.
6.	Agarose gel electrophoresis (AGE)	Exposed to H_2_O_2_ to induce oxidation resulted in a slight difference in migration	Same as Moderna; there is a slight difference in migration	No evidence of degradation.
7.	Effect of thermal stress. Size distribution after 4 and 8 days at 40 °C and 60% RH. (Study the tendency to aggregate and degrade)	Seen to be more resilient to physical instability than the Pfizer vaccine. The same mean size and PDI values all over the test. (187.9 ± 1.735 nm and 0.181 ± 0.01546 unstressed sample vs. 182.3 ± 1.735 nm and 0.174 ± 0.01275 on 8th day)	Experienced growth of bigger population (approximately 1000 nm), explaining the slight increase in Z-average (86.35 ± 0.5841 nm and 0.2141 ± 0.0058 for unstressed sample vs. 96.92 ± 0.8718 nm and 0.2495 ± 0.0055 after 8 days under thermal stress)	Moderna showed longer stability as compared to the Pfizer vaccine at room temperature.
8.	Internalization of SLN in A549 cells.	Each cell contains higher and more internalization. Moderna has a stronger ability to invade cytoplasm. More particles attach to the membrane as compared to the one undergoing internalization.	Uptake was slow. Reaches mostly to cells whereas few attach to the membrane	The uptake of Moderna SLN was more sustained concerning Pfizer due to the restricted availability of endosomal vesicles, delaying the continuous flow of NPs from the membrane to the cytoplasm.

**Table 4 pathogens-13-00500-t004:** Clinical trials for mRNA vaccines against influenza (as registered at ClinicalTrials.gov, accessed on 30 May 2024). Full information on each vaccine, including the sponsor and the outcomes, can be found by referring to the study URL as per the table.

NCT Number	Study Title	Study URL	Study Status	Interventions, All Biologicals, Unless Stated Otherwise	Phases	Enrollment
NCT05827926	A Study of mRNA-based Influenza and SARS-CoV-2 (COVID-19) Multi-component Vaccines in Healthy Adults	https://clinicaltrials.gov/study/NCT05827926 (accessed on 30 May 2024)	Active, not recruiting	Influenza Vaccine 1|mRNA-1083.1|mRNA-1083.2|mRNA-1083.3|Investigational Influenza Vaccine 1|Investigational COVID-19 Vaccine 1|COVID-19 Vaccine 1|Investigational Influenza Vaccine 2|Influenza Vaccine 2|mRNA-1083|Investigational COVID-19 Vaccine 2|COVID-19 Vaccine 2	PHASE1|PHASE2	1763
NCT06125691	Safety and Immunogenicity First-in-human Dose-ranging Study of Self-Amplifying RNA Seasonal Influenza Vaccine in Adults	https://clinicaltrials.gov/study/NCT06125691 (accessed on 30 May 2024)	Recruiting	ARCT-2138|Licensed Quadrivalent Vaccine for younger adults|Licensed Quadrivalent Vaccine for older adults	PHASE1	132
NCT05823974	A Study to Assess the Safety and Immune Response of a Vaccine Against Influenza in Healthy Younger and Older Adults	https://clinicaltrials.gov/study/NCT05823974 (accessed on 30 May 2024)	Active, not recruiting	Flu mRNA|COMBINATION_PRODUCT: Control 1|COMBINATION_PRODUCT: Control 2	PHASE1|PHASE2	1256
NCT05868382	Study to Evaluate the Safety, Reactogenicity, and Immunogenicity of mRNA Vaccine Candidate Variations in Healthy Adults	https://clinicaltrials.gov/study/NCT05868382 (accessed on 30 May 2024)	Completed	mRNA-1010|mRNA-1010.4|mRNA-1010.6	PHASE2	270
NCT05566639	A Study of mRNA-1010 Seasonal Influenza Vaccine in Adults 50 Years Old and Older	https://clinicaltrials.gov/study/NCT05566639 (accessed on 30 May 2024)	Completed	mRNA-1010|Licensed quadrivalent inactivated seasonal influenza vaccine	PHASE3	22,510
NCT05553301	Safety and Immunogenicity of Quadrivalent Influenza mRNA Vaccine MRT5407 in Adult Participants18 Years of Age and Older	https://clinicaltrials.gov/study/NCT05553301 (accessed on 30 May 2024)	Completed	Quadrivalent Influenza mRNA Vaccine MRT5407|Quadrivalent Recombinant Influenza Vaccine|Quadrivalent Influenza Standard Dose Vaccine|Quadrivalent Influenza High-Dose Vaccine	PHASE1|PHASE2	560
NCT05446740	A Study on the Safety, Reactogenicity and Immune Response of a Vaccine Against Influenza in Healthy Younger and Older Adults	https://clinicaltrials.gov/study/NCT05446740 (accessed on 30 May 2024)	Completed	GSK4382276A Dose level 1|GSK4382276A Dose level 2|GSK4382276A Dose level 3|GSK4382276A Dose level 4|GSK4382276A Dose level 6|GSK4382276A Dose level 7|GSK4382276A Dose level 8|GSK4382276A Dose level 9|GSK4382276A Dose level 10|GSK4382276A Dose level 11|COMBINATION_PRODUCT: FDQ21A-NH|COMBINATION_PRODUCT: FDQ22A-NH	PHASE1	324
NCT06097273	A Study of mRNA-1083 (SARS-CoV-2 and Influenza) Vaccine in Healthy Adult Participants, ≥50 Years of Age	https://clinicaltrials.gov/study/NCT06097273 (accessed on 30 May 2024)	Active, not recruiting	mRNA-1083|Placebo|Influenza Vaccine|COVID-19 Vaccine	PHASE3	8075
NCT05755620	A Study to Evaluate the Safety and Immunogenicity of a Single Dose of H1ssF-3928 mRNA-LNP in Healthy Adults	https://clinicaltrials.gov/study/NCT05755620 (accessed on 30 May 2024)	Recruiting	Influenza Virus Quadrivalent Inactivated Vaccine|OTHER: Sodium Chloride, 0.9%|VRC-FLUNPF099-00-VP (H1ssF_3928)	PHASE1	50
NCT05426174	Study to Assess the Safety and Immunogenicity of Monovalent mRNA NA Vaccine in Adult Participants 18 Years of Age and Older	https://clinicaltrials.gov/study/NCT05426174 (accessed on 30 May 2024)	Completed	mRNA NA vaccine|High Dose Quadrivalent Influenza Vaccine	PHASE1	233
NCT05397223	A Study of Modified mRNA Vaccines in Healthy Adults	https://clinicaltrials.gov/study/NCT05397223 (accessed on 30 May 2024)	Active, not recruiting	mRNA-1273|mRNA-1010|mRNA-1345|FLUADÂ^®^|mRNA-1647	PHASE1	308
NCT05252338	A Study to Evaluate the Safety, Reactogenicity and Immunogenicity of Vaccine CVSQIV in Healthy Adults	https://clinicaltrials.gov/study/NCT05252338 (accessed on 30 May 2024)	Completed	CVSQIV	PHASE1	240
NCT03345043	Safety, Tolerability, and Immunogenicity of VAL-339851 in Healthy Adult Subjects	https://clinicaltrials.gov/study/NCT03345043 (accessed on 30 May 2024)	Completed	VAL-339851|OTHER: Placebo	PHASE1	156
NCT06118151	Safety and Immunogenicity of a Monovalent mRNA Vaccine Encoding Influenza Hemagglutinin in Adult Participants 18 Years of Age and Older	https://clinicaltrials.gov/study/NCT06118151 (accessed on 30 May 2024)	Completed	Influenza Hemagglutinin mRNA vaccine|Quadrivalent Recombinant Influenza Vaccine	PHASE1	388
NCT05650554	Safety and Immunogenicity of Quadrivalent Influenza mRNA Vaccine MRT5413 in Adult Participants18 Years of Age and Older	https://clinicaltrials.gov/study/NCT05650554 (accessed on 30 May 2024)	Completed	Quadrivalent Influenza mRNA Vaccine MRT5413|Quadrivalent Recombinant Influenza Vaccine|Quadrivalent Influenza Standard Dose Vaccine|Quadrivalent Influenza High-Dose Vaccine	PHASE1|PHASE2	682
NCT05624606	Safety and Immunogenicity of Quadrivalent Influenza mRNA Vaccine MRT5410 in Adult Participants 18 Years of Age and Older	https://clinicaltrials.gov/study/NCT05624606 (accessed on 30 May 2024)	Completed	Quadrivalent Influenza mRNA Vaccine MRT5410|Quadrivalent Recombinant Influenza vaccine RIV4|Quadrivalent Inactivated Influenza Standard Dose QIV-SD|Quadrivalent Inactivated Influenza High Dose QIV-HD	PHASE1|PHASE2	682
NCT06361875	A Study to Investigate the Safety and Immunogenicity of the Quadrivalent Influenza mRNA Vaccines in Adults Aged 18 Years and Above	https://clinicaltrials.gov/study/NCT06361875 (accessed on 30 May 2024)	Recruiting	Quadrivalent Influenza mRNA Vaccine MRT5421|Quadrivalent Influenza mRNA Vaccine MRT5424|Quadrivalent Influenza mRNA Vaccine MRT5429|Quadrivalent Influenza Standard Dose Vaccine|Quadrivalent Influenza High-Dose Vaccine|Quadrivalent Recombinant Influenza Vaccine	PHASE1|PHASE2	1002
NCT05945485	A Study to Evaluate the Safety and Immunogenicity of Two Doses of DCVC H1 HA mRNA-LNP in Healthy Adults	https://clinicaltrials.gov/study/NCT05945485 (accessed on 30 May 2024)	Recruiting	DCVC H1 HA mRNA vaccine|Quadrivalent Recombinant Seasonal Influenza Vaccine|OTHER: Sodium Chloride, 0.9%	PHASE1	50
NCT03076385	Safety, Tolerability, and Immunogenicity of VAL-506440 in Healthy Adult Subjects	https://clinicaltrials.gov/study/NCT03076385 (accessed on 30 May 2024)	Completed	VAL-506440|OTHER: Placebo	PHASE1	201
NCT06028347	Safety, Reactogenicity, and Immunogenicity Study of a Self-Amplifying mRNA Influenza Vaccine in Healthy Adults	https://clinicaltrials.gov/study/NCT06028347 (accessed on 30 May 2024)	Active, not recruiting	sa-mRNA vaccine Dose 1|sa-mRNA vaccine Dose 2|sa-mRNA vaccine Dose 3|Placebo	PHASE1	96
NCT05827978	Study of mRNA-1010 Seasonal Influenza Vaccine in Adults	https://clinicaltrials.gov/study/NCT05827978 (accessed on 30 May 2024)	Active, not recruiting	mRNA-1010|Licensed Quadrivalent Inactivated Seasonal Influenza Vaccine	PHASE3	8400
NCT06431607	A Study to Find the Dose and Assess the Immune Response and Safety of a Vaccine Against Influenza in Healthy Younger and Older Adults	https://clinicaltrials.gov/study/NCT06431607 (accessed on 30 May 2024)	Recruiting	Flu Seasonal mRNA Formulation 1|Flu Seasonal mRNA Formulation 2|Flu Seasonal mRNA Formulation 3|Flu Seasonal mRNA Formulation 4|Flu Seasonal mRNA Formulation 5|Flu Seasonal mRNA Formulation 6|Flu Seasonal mRNA Formulation 7|Flu Seasonal mRNA Formulation 8|COMBINATION_PRODUCT: Active Comparator 1|COMBINATION_PRODUCT: Active Comparator 2	PHASE2	500
NCT06382311	A Study to Find and Confirm the Dose and Assess Safety, Reactogenicity and Immune Response of a Vaccine Against Pandemic H5N1 Influenza Virus in Healthy Younger and Older Adults	https://clinicaltrials.gov/study/NCT06382311 (accessed on 30 May 2024)	Recruiting	Flu Pandemic mRNA_Dose level 1|Flu Pandemic mRNA_Dose level 2|Flu Pandemic mRNA_ Dose level 3.|Flu Pandemic mRNA_ Dose level 4|Flu Pandemic mRNA_Dose level 5|Flu Pandemic mRNA_Dose level 6|Flu Pandemic mRNA_Dose level 7|DRUG: Placebo	PHASE1|PHASE2	1080
NCT05829356	Substudy 01—Safety and Immunogenicity of One Monovalent Modified mRNA Vaccine Encoding Influenza Hemagglutinin With LNP, in Adult Participants Aged 18 to 49 Years and 60 Years and Above	https://clinicaltrials.gov/study/NCT05829356 (accessed on 30 May 2024)	Completed	H3 mRNA / LNP Vaccine|Quadrivalent recombinant influenza Vaccine (RIV4)	PHASE1	159
NCT05585632	A Safety, Reactogenicity, and Immunogenicity Study of mRNA-1045 (Influenza and Respiratory Syncytial Virus [RSV]) or mRNA-1230 (Influenza, RSV, and Severe Acute Respiratory Syndrome Coronavirus 2 [SARS-CoV-2]) Vaccine in Adults 50 to 75 Years Old	https://clinicaltrials.gov/study/NCT05585632 (accessed on 30 May 2024)	Completed	mRNA-1010|mRNA-1345|mRNA-1273.214|mRNA-1045|mRNA-1230	PHASE1	392

## Data Availability

Not Applicable.

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
