# Peer review of "Brief Insights into mRNA Vaccines: Their Successful Production and Nanoformulation for Effective Response against COVID-19 and Their Potential Success for Influenza A and B"

_pathogens, 2024, doi:10.3390/pathogens13060500_

Round 1

Reviewer 1 Report

Comments and Suggestions for Authors

This is a detailed review on an interesting topic.

Since the technology of mRNA vaccines has not been established for influenza vaccines and is not in use for this type of vaccine I suggest to rephrase the title and clearly state that in the text.

The paper mainly addresses  pharmacological properties of mRNA vaccines. It could be of use to elaborate their immunological correlates as well.

Language editing is needed throughout the paper.

Comments on the Quality of English Language

English language editing is required

Author Response

This is a detailed review on an interesting topic.

Response: Thanks for the comment

Since the technology of mRNA vaccines has not been established for influenza vaccines and is not in use for this type of vaccine I suggest to rephrase the title and clearly state that in the text.

Response: Done, please refer to the green highlights in the title and in the text as a reflection on this point.

The paper mainly addresses pharmacological properties of mRNA vaccines. It could be of use to elaborate their immunological correlates as well.

Response: A paragraph 

Language editing is needed throughout the paper.

Response: Done

Reviewer 2 Report

Comments and Suggestions for Authors

The review article discusses the production, formulation, and stabilization of mRNA vaccines, with a special emphasis on SARS-CoV-2 and the Influenza virus. The manuscript is comprehensive and well-organized; however, it feels more like reading a book chapter than a review article. Almost all the figures and tables are from previously published articles, and the authors' innovativeness is missing. The authors provide a detailed description of various aspects of mRNA vaccines, such as their structure, chemistry, history, principles, stability, synthesis, and challenges. All subtopics are clear and elaborate, but many articles cover this information similarly. The section on the "Application of mRNA vaccines for COVID-19 and influenza viruses" is interesting and informative. It would highly attract readers if the authors included a table about the clinical trials of mRNA vaccines against flu and COVID-19.

I suggest that the authors reframe the manuscript slightly to attract readers

Author Response

The review article discusses the production, formulation, and stabilization of mRNA vaccines, with a special emphasis on SARS-CoV-2 and the Influenza virus. The manuscript is comprehensive and well-organized;

Response: Thanks for the comment.

however, it feels more like reading a book chapter than a review article.

Response: The titles and subtitles have been arranged to make the manuscript readable in a sequence manner from isolation, purification, and modification of mRNA to its formulation in different particulate forms.

Almost all the figures and tables are from previously published articles, and the authors' innovativeness is missing.

Response: Table 3 is constructed by the authors, using the data from a specific article which is cited within the title of the table.

The authors provide a detailed description of various aspects of mRNA vaccines, such as their structure, chemistry, history, principles, stability, synthesis, and challenges. All subtopics are clear and elaborate,

Response: Thanks for the comment

but many articles cover this information similarly.

Response: However, in our manuscript we covered both influenza and Covid-19 LNP-mRNA vaccines, as there are still challenges to formulate LNP-mRNA vaccine for different subtypes of Influenza A.

The section on the "Application of mRNA vaccines for COVID-19 and influenza viruses" is interesting and informative.

Response: Thanks for the comment.

It would highly attract readers if the authors included a table about the clinical trials of mRNA vaccines against flu and COVID-19.

Response: Two tables are included, Tables 2 and 4; the changes/additions are yellow highlighted.

I suggest that the authors reframe the manuscript slightly to attract readers

Response: Done, please refer to the green and yellow highlights within the manuscript.

Round 2

Reviewer 1 Report

Comments and Suggestions for Authors

No further comments

Comments on the Quality of English Language

improved

Author Response

No further comments

Response: Many Thanks

Reviewer 2 Report

Comments and Suggestions for Authors

For clinical trials table authors can include more information like phase , population, sample size, outcome and results etc.

Author Response

For clinical trials table authors can include more information like phase, population, sample size, outcome and results etc.

Response: More information (with blue highlights) was added to the tables of clinical studies. Results for most studies are not available yet, hence, a phrase was added into the text "blue highlighted" to reflect referring to the study URL for any other information needed.